# TrackMamba: Mamba-Transformer Tracking

## Abstract

Current one-stream Transformer-based trackers are quality but unfriendly to memory consumption of large resolution and long sequence, both of which are crucial keys to tracking tasks. Recently structured state space model (SSM) demonstrates promising performance and efficiency in sequence modeling but struggles to retrieve due to the limited hidden state number. To solve the computation challenge and explore the potential of Mamba, we propose TrackMamba, a Mamba-Transformer tracker containing TrackMamba Blocks and Attention Blocks. In order to better harness the scanning in TrackMamba Blocks for inter- and intra-frame modeling, we introduce various scan patterns for rearrangement and flipping. Furthermore, we propose Target Enhancement, including Temporal Token for target aggregation and search enhancement, and Temporal Mamba for target information cross-frame propagation. Extensive experiments show TrackMamba performs better than the first-generation one-stream Transformer-based tracker at same resolution and mitigates consumption growth when enlarging resolution, exhibiting the potential of Mamba-based model for large-resolution tracking.

## 1 Introduction

Visual object tracking aims to locate the target object in video sequences based on its initial state, which is one of the fundamental tasks in computer vision. Except for the traditional challenges that most of the work endeavors to solve, such as object deformations, occlusion, and confusion with similar objects, there are also challenges related to model efficiency including computation and memory burden when enlarging input resolution and extending long sequence so that most trackers work on small resolution for large-resolution datasets, resulting in inadequate performance.

Prevailing trackers follow a three-stage pipeline that extracts features of template and search region separately, then models their cross-relations, and predict box finally, such as Siamese-based tracker (Bertinetto et al., 2016; Li et al., 2019). With the help of Transformer (Vaswani et al., 2017), several trackers (Chen et al., 2021; Yan et al., 2021) adopt Attention to enhance cross-relation. Most of the recently proposed Transformer-based trackers (Cui et al., 2024; Ye et al., 2022) are changed to one-stream pipeline for joint feature learning and relation modeling inside backbone which obtains better target-relevant search features. Thanks to the strong global modeling ability of Transformer and their well-pretrained backbone (He et al., 2022), they have achieved remarkable success. However, the quadratic complexity of attention faces challenges of computation burden when enlarging the image resolution which is critical for spatial modeling and location. Meanwhile, several works focused on temporal modeling to handle appearance changes and distractor, such as additional dynamic online templates (Song et al., 2023; Cui et al., 2024) or progressively learnable tokens (Shi et al., 2024; Zheng et al., 2024). Unfortunately, directly extending the temporal length introduces significant computation to Transformer-based methods still due to the quadratic complexity.

On the other hand, structured state space models (SSMs) (Gu et al., 2022a) can model sequences with linear complexity, demonstrating robust performance across a spectrum of sequence modeling tasks while maintaining efficiency. Selective State Space Model (S6), a variant of SSMs, also known as Mamba (Gu & Dao, 2023), has garnered significant attention within the vision community and demonstrated comparable performance to Transformer (Vaswani et al., 2017) across numerous vision tasks. Selective Scan Mechanism, as the core operation of Mamba, makes SSM parameters data-dependent from input sequence, which enhances context-aware sensitivity. With the ability of relevant context selection and linear complexity, it is natural to employ Mamba to address the pres-

sures from mentioned problem of resolution expanding and sequence growth. To the best of our knowledge, Mamba remains untouched for single object tracking task.

Although SSM behaves well in sequences modeling, several studies (Park et al., 2024; Wen et al., 2024; Pantazopoulos et al., 2024) have demonstrated that pure Mamba inherently lacks the ability of retrieval and localization (Wen et al., 2024; Pantazopoulos et al., 2024) due to the limited hidden state number and suggest to incorporate attention as hybrid model to overcome the limitations.

To solve the mentioned computation challenge and explore the potential of Mamba in single object tracking tasks, we propose **TrackMamba**, a novel Mamba-based tracker with great performance both on tracking accuracy and memory consumption. Specifically, inspired by above studies, we adopt MambaVision (Hatamizadeh & Kautz, 2024), a hybrid Mamba-Transformer model, as our backbone for promising performance while maintaining efficiency. We first introduce TrackMamba Block as the core design of tracker, which performs both feature extraction and interaction with scanning. In addition, considering the rearrangement and flipping of input sequence play a critical role in scanning, we discuss them in detail and propose various Scan Patterns that reasonably solve the information sources and disturbances problem during scanning. Furthermore, to complement the lack of direct cross-frame interaction for Mamba scanning, we introduce Target Enhancement, including Temporal Token that performs target aggregation and search feature enhancement, and Temporal Mamba for target information propagation by transferring along these tokens with Mamba.

Extensive experiments on several benchmarks demonstrate our TrackMamba performs better than the first-generation one-stream tracker (Cui et al., 2024) with the same resolution. When scaling up the resolution, our tracker has a strong improvement on large resolution benchmarks, such as GOT-10k (Huang et al., 2021), and mitigates computational consumption. Moreover, the current framework has untapped potential due to the limitations of the backbone and pre-training. We believe that with a better backbone, it could be scaled to higher resolutions better to improve performance.

Our main contributions are summarized as follows:

1. We propose a novel tracking framework, termed as TrackMamba, which adopts the hybrid Mamba-Transformer model and enables accurate and low-consumption tracking.

2. For better scanning input sequence in TrackMamba Block, we introduce various Scan Patterns to arrange and flip them, solving the source and disturbances problem.

3. We propose Target Enhancement, ina Temporal Token for target feature aggregation and refinement, and Temporal Mamba for modeling them, enabling information highly propagation across frames.

4. Extensive experiments on multiple benchmarks show better performance of our tracker than the first-generation one-stream tracker at the same image resolution while demonstrating performance growth and lower consumption at larger resolution.

## 2 RELATED WORK

### 2.1 MAMBA IN VISION

The State Space Model (SSM) (Gu et al., 2022a) can model sequences with linear complexity, and Mamba (Gu & Dao, 2023) introduces a novel data-dependent parametrization approach and presents an efficient hardware-aware algorithm based on selective scan, achieving comparable performance and better efficiency to Transformers in language modeling of long sequence NLP tasks.

Recently, Mamba, with its linear complexity in long-range modeling, has been introduced to many visual tasks and demonstrated promising performance. Vim (Zhu et al., 2024) constructs a ViT-like (Dosovitskiy et al., 2021) vision backbone with Mamba. VMamba (Liu et al., 2024) proposes a hierarchical vision model based on Mamba with four-directional scanning. VideoMamba (Li et al., 2024) leverages the linear-complexity operator inherent in Mamba to overcome the challenges of the dual challenges of local redundancy and global dependencies in video data. This success has led to its adoption in subsequent tasks, such as generation (Teng et al., 2024), point cloud analysis(Zhang et al., 2024b; Liang et al., 2024), image restoration (Guo et al., 2024), video frame interpolation (Zhang et al., 2024a), medical image segmentation (Ma et al., 2024; Wang et al., 2024b).

## 2.2 HYBRID MODEL

Despite the sequence modeling ability of State Space Model with linear complexity, its retrieval capacity is limited by relying on the finite number of internal states (Park et al., 2024; Wen et al., 2024; Jelassi et al., 2024), which results in suboptimal performance across various tasks, such as multi-query associative recall (MQAR) task (Park et al., 2024) and visual grounding (Pantazopoulos et al., 2024). To mitigate this issue, some research has focused on efficiently increasing the number of internal states (Dao & Gu, 2024; Qin et al., 2024) or refining the update rules (Schlag et al., 2021).

Beyond above studies, more works explored to insert attention mechanisms in Mamba (Park et al., 2024; Wen et al., 2024; Waleffe et al., 2024) to explore hybrid models, yielding strong performance across various tasks, such as language modeling (Lieber et al., 2024), image classification (Hatamizadeh & Kautz, 2024), point cloud (Wang et al., 2024a), and image generation (Fei et al., 2024). This trend highlights the great potential of hybrid architectures across diverse applications. Based on the trend, we additionally found similar formalization of the MQAR and tracking tasks and therefore chose the Mamba-based hybrid model for single object tracking task.

## 2.3 SINGLE OBJECT TRACKING

Classics trackers follow a three-stage architecture, including separate feature extraction of template and search frames, integration between them, and target location with a box head. Siamese-based trackers (Bertinetto et al., 2016; Li et al., 2019) adopt a correlation operation to model the appearance similarity and correlation. Based on the success of Transformer (Vaswani et al., 2017), some trackers, such as TransT (Chen et al., 2021) and STRAK (Yan et al., 2021) adopt attention to capture the global correlation while stilling following the three-stage architecture. In contrast, MixFormer (Cui et al., 2024) performs both feature extraction and interaction within the Transformer-based backbone as a representative of the first one-stream generation. Despite their great performance, the quadratic computational complexity of self-attention has hindered the development of long-range modeling and large sizes, while both of them play a key role in tracking.

In fact, Mamba has already made a mark in other tracking tasks. For instance, several works (Huang et al., 2024a; Xiao et al., 2024; Hu et al., 2024) leverage Mamba as a motion predictor to model trajectories in multi-object tracking, MambaVT (Lai et al., 2024) jointly model RGB and TIR with trajectories in RGB-T object tracking, and MambaFETrack (Huang et al., 2024b) adopt Mamba to modality interaction with event streams in RGB-Event tracking. In contrast, our work is the first to investigate the application of the Mamba-based backbone model for single object tracking tasks.

## 3 PRELIMINARIES

**State Space Models and Mamba.** State Space Models (SSMs) (Gu et al., 2022b) are based on continuous systems that map a 1D continuous input sequence $x(t) \in \mathbb{R}$ to an output $y(t) \in \mathbb{R}$ via a learnable hidden state $h(t) \in \mathbb{R}^N$ for a state size $N$, parameterized by $\boldsymbol{A} \in \mathbb{R}^{N \times N}$ as the evolution parameter, $\boldsymbol{B} \in \mathbb{R}^{1 \times N}$ and $\boldsymbol{C} \in \mathbb{R}^{1 \times N}$ as the projection parameters, which typically formulated as following linear ordinary differential equations (ODEs):

$$
\begin{aligned}
h'(t) &= \boldsymbol{A}h(t) + \boldsymbol{B}x(t), \\
y(t) &= \boldsymbol{C}h(t),
\end{aligned}
\tag{1}
$$

With a timescale parameter $\Delta$, the continuous parameters $\boldsymbol{A}$, $\boldsymbol{B}$ could be discretized to discrete parameters $\bar{\boldsymbol{A}}$, $\bar{\boldsymbol{B}}$ according to the zero-order hold (ZOH) rule, which can be formulated as:

$$
\begin{aligned}
\bar{\boldsymbol{A}} &= \exp(\Delta\boldsymbol{A}), \\
\bar{\boldsymbol{B}} &= (\Delta\boldsymbol{A})^{-1}(\exp(\Delta\boldsymbol{A}) - \boldsymbol{I}) \cdot (\Delta\boldsymbol{B}),
\end{aligned}
\tag{2}
$$

Thus, the Eq.1 can be expressed with discrete parameters to a recurrent formulation as:

$$
\begin{aligned}
h(t) &= \bar{\boldsymbol{A}}h(t-1) + \bar{\boldsymbol{B}}x(t), \\
y(t) &= \boldsymbol{C}h(t),
\end{aligned}
\tag{3}
$$

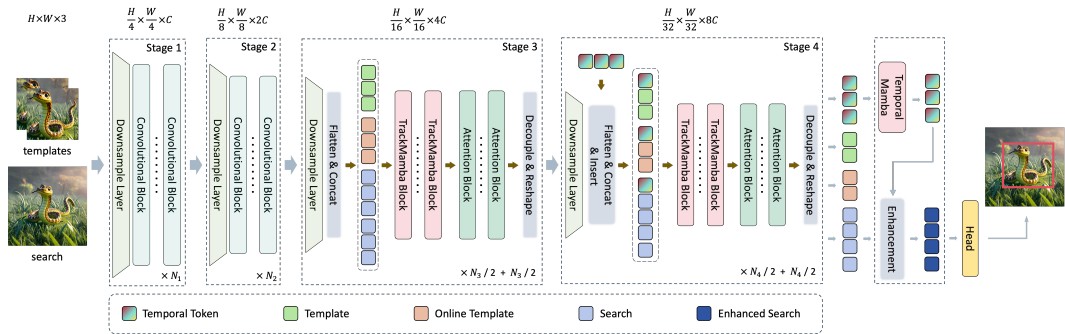

Figure 1: The overview framework of the proposed TrackMamba.

In contrast to traditional models that rely heavily on linear time-invariant SSMs, Mamba (Gu & Dao, 2023) extends the SSM by introducing Selective Scan Mechanism (S6) as its core operator. With S6 operation, three linear projection layers $S_\Delta(x), S_B(x), S_C(x)$ are introduced to directly derived the parameter $\boldsymbol{B} \in \mathbb{R}^{L \times N}$, $\boldsymbol{C} \in \mathbb{R}^{L \times N}$, and $\Delta \in \mathbb{R}^{L \times N}$ from the input data $x(t) \in \mathbb{R}^{L \times N}$ for data-dependent processing in Eq.2 which enhances its context-aware sensitivity. Additionally, Mamba also presents an efficient hardware-aware implementation.

**Formulate Tracking as MQAR.** Multi-query associative recall (MQAR) (Park et al., 2024) task provides a sequence of query $\{q_1, q_2, \ldots, q_m\}$ and key-value pairs $\{(k_1, v_1), (k_2, v_2), \ldots, (k_n, v_n)\}$. For each query $q_j$, there exist some keys that satisfy $q_j = k_l$, and the model needs to recall $v_l$ for each query, producing $m$ outputs total. Tracking can be formulated as an MQAR problem by treating the template images as key-value pairs and the search image as a set of query tokens. Beyond this, tracking introduces unique challenges, such as appearance variations, occlusion, and distractors, requiring more robust matching.

**Tracking Challenging Pure SSMs.** Due to the limited hidden state dimension for carrying information, several studies (Park et al., 2024; Wen et al., 2024; Jelassi et al., 2024; Waleffe et al., 2024) indicate that SSMs struggle to accurately retrieve the vectors in MQAR task and are overwhelmed if the context increases substantially, which leads to a lack of retrieval capabilities for matching-based task, such as localization (Pantazopoulos et al., 2024) and tracking. To address this, they introduced attention mechanisms to yield a hybrid model. Inspired by these efforts, we adopt the hybrid framework, MambaVision (Hatamizadeh & Kautz, 2024) rather than the pure Mamba model, so as to unleash the strong power of the Mamba-based model in preserving sufficient target information and integrating it into the search.

## 4 METHOD

In this section, we describe our proposed tracker, TrackMamba. First, we begin with an overview description of the framework. Then, we propose the core TrackMamba Block, which replaces the Attention Block of one-stream transformer-based trackers, enabling the search features to be more consistent with the target in addition to feature extraction. Furthermore, since the arrangement and flipping strategies of the input sequences are critical of Mamba scanning, we give a detailed discussion on scanning patterns and introduce three various patterns in TrackMamba Block. In addition, we present Target Enhancement, containing Temporal Token for target feature aggregation and enhancing target-relevant search, and Temporal Mamba for target information cross-frame propagation within one token for each frame. Finally, we describe the training and inference of TrackMamba.

### 4.1 OVERVIEW

As shown in Fig. 1, the input of tracker contains $T$ templates $\boldsymbol{z} \in \mathbb{R}^{T \times 3 \times H_z \times W_z}$ and search region $\boldsymbol{x} \in \mathbb{R}^{\times H_x \times W_x}$. They are first downsampled to $\frac{1}{4}$ and $\frac{1}{8}$ scale with the first two convolutional stages. At the beginning of the next two stages, they are downsampled $\frac{1}{16}$ or $\frac{1}{32}$ scale, divided and flatten to token sequence $\boldsymbol{z_p} \in \mathbb{R}^{T \times N_z \times (C \cdot P^2)}$ and $\boldsymbol{x_p} \in \mathbb{R}^{N_x \times (C \cdot P^2)}$, where $C$ and $P$ are

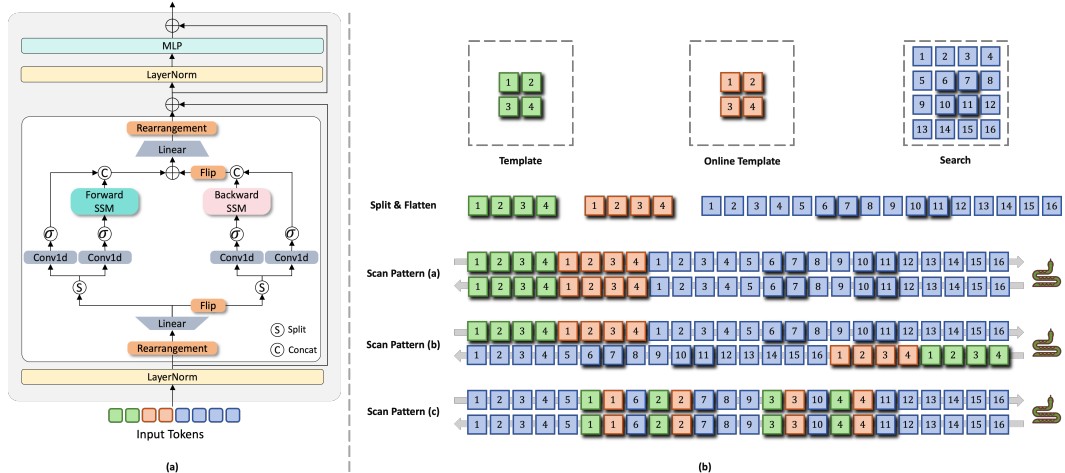

Figure 2: **(a)** The Detail structure of TrackMamba Block, wherein the input sequences are rearranged and flipped in different ways followed by parallel forward and backward scanning. **(b)** Three variants of **Scan Patterns**, including rearrangements and flip strategy which both profoundly impact sequence modeling. The first row of each Scan Pattern represents a forward scan and the second row represents a parallel backward scan. The shaded tokens are located the same center position of target in templates and search during the cropping process of tracking.

the channel and patch resolution in this stage, $N_z = H_z W_z / P^2$ and $N_x = H_x W_x / P^2$ are patch number of templates and search. The template sequence $\boldsymbol{E_z^0} \in \mathbb{R}^{T \times N_z \times D}$ and search sequence $\boldsymbol{E_x^0} \in \mathbb{R}^{N_x \times D}$ are concatenated as $\boldsymbol{E_{zx}^0} = [\boldsymbol{E_z^0}; \boldsymbol{E_x^0}]$. They are then fed, along with Temporal Tokens $\boldsymbol{E_c^0} \in \mathbb{R}^{(T+1) \times 1 \times D}$, into TrackMamba and Attention Blocks, allowing for simultaneous feature extraction and target-search integration, while aggregating online targets information into the Temporal Tokens. After the last two stages, we decouple it into template $\boldsymbol{E_z^L}$, search $\boldsymbol{E_x^L}$ and Temporal Tokens $\boldsymbol{E_c^L}$, and input Temporal Tokens into Temporal Mamba for temporal modeling cross-frames. Finally, the search region are refined with its Temporal Token as $\bar{\boldsymbol{E}}_x^L$, re-shaped to a 2D feature map, and the regression head directly adopts this target-relevant search features together multi-scale feature from backbone for box prediction.

## 4.2 TRACKMAMBA BLOCK

As illustrated in Fig. 1, each stage contains a set of TrackMamba Blocks and Attention Blocks. In TrackMamba Block, as shown in Fig. 2(a), the concatenated sequences are first re-arranged and flipped with various rearrangement and inversion strategies which we describe in detail later. Then we feed the two sequences to Forward SSM and Backward SSM separately. Finally, we flip back, add them together, and re-arrange back before passing them to the next block:

$$\bar{\boldsymbol{E}}_{zx,\text{forward}}^l = Rearrange(\boldsymbol{E}_{zx}^l),$$
$$\bar{\boldsymbol{E}}_{zx,\text{backward}}^l = Flip(\bar{\boldsymbol{E}}_{zx}^l),$$
$$\bar{\boldsymbol{y}}_{zx,\text{forward}}, \bar{\boldsymbol{y}}_{zx,\text{backward}} = SSM_{\text{forward}}(\bar{\boldsymbol{E}}_{zx,\text{forward}}^l), SSM_{\text{backward}}(\bar{\boldsymbol{E}}_{zx,\text{backward}}^l), \quad (4)$$
$$\bar{\boldsymbol{E}}_{zx}^{l+1} = \bar{\boldsymbol{y}}_{zx,\text{forward}} + Flip_{\text{back}}(\bar{\boldsymbol{y}}_{zx,\text{backward}}),$$
$$\boldsymbol{E}_{zx}^{l+1} = Rearrange_{\text{back}}(\bar{\boldsymbol{E}}_{zx}^{l+1}),$$

where $\bar{\boldsymbol{E}}_{zx}$ and $\bar{\boldsymbol{y}}_{zx}$ are rearranged sequences. During the scanning template in SSM, the target information of the template is injected into the hidden state, which makes the hidden state gradually become target-relevant and also see multiple target forms across various templates. After that, it begins to scan the search area, making it receive target information from the hidden state and thus becomes relevant to the target in order to locate the target. Note that our proposed TrackMamba Block adopt scanning mechanism with linear complexity rather than quadratic complexity of attention, which is convenient for long sequences and large sizes that are critical for tracking task.

As for Attention Block, we adopt Asymmetric Mixed attention module of MixFormer (Cui et al., 2024) for simultaneously extraction and interaction. It removes the unnecessary target-to-search cross-attention which remains template token unchanged by search as follows:

$$Q_z, K_z, V_z = W_Q \boldsymbol{E}_{\boldsymbol{z}}^{\boldsymbol{l}}, W_K \boldsymbol{E}_{\boldsymbol{z}}^{\boldsymbol{l}}, W_V \boldsymbol{E}_{\boldsymbol{z}}^{\boldsymbol{l}}; \quad Q_x, K_x, V_x = W_Q \boldsymbol{E}_{\boldsymbol{x}}^{\boldsymbol{l}}, W_K \boldsymbol{E}_{\boldsymbol{x}}^{\boldsymbol{l}}, W_V \boldsymbol{E}_{\boldsymbol{x}}^{\boldsymbol{l}},$$
$$\boldsymbol{E}_{\boldsymbol{z}}^{\boldsymbol{l+1}} = Softmax(\frac{Q_z K_z}{\sqrt{d_k}}) \cdot V_z; \quad \boldsymbol{E}_{\boldsymbol{x}}^{\boldsymbol{l+1}} = Softmax(\frac{Q_x [K_z; K_x]}{\sqrt{d_k}}) \cdot [V_z; V_x], \quad (5)$$

where $W_Q$, $W_K$, and $W_V$ are the matrix of Attention.

**Discussion on Scan Patterns.** The Scan Pattern of input sequence, including rearrangement and flipping, determines the transfer flow, and it is essential to analyze which Scan Pattern is more suitable for intra- and inter-frame modeling. First, if the sequence simply flipped as a whole into search-templates sequence, the backward scan begins with search region that contains lots of background while ends with templates that are actually needed as source. This violates the purpose of the transfer and disturbs the template features. Next, based on the cropping process of tracking, the object is always at the center of the cropped-out frame and the search region has twice scale factor than templates. It can be assumed that the object is at almost the same position across, i.e., the shaded tokens in Fig. 2. We could interleaved rearrange tokens with the same position for direct inter-frame modeling. In summary, we propose three various Scan Patterns shown in Fig. 2 as follows:

(a) sequential-rearrange-whole-flip: Flipping the sequential sequence as a whole sequence,

(b) sequential-rearrange-separate-flip: Replacing the whole flipping with separate flipping to fix the source and disturbance problems, and keeping the sequential order unchanged,

(c) interleaved-rearrange-whole-flip: Interleaved rearranging the tokens in the same center position of different frames and keep the position of the background tokens on the periphery of the search region unchanged, generating a splice sequence for direct inter-frame modeling.

As the critical pole for scanning, these operations strongly affect the model performance. Our experiments, in Section 5.3, will verify the performance and analyze them in further detail.

### 4.3 TARGET ENHANCEMENT WITH TEMPORAL TOKEN AND TEMPORAL MAMBA

Admittedly, Mamba enables target delivery with its long sequence capability while still lacking direct cross-frame modeling. Inspired by class token in image classification, which aggregates object feature, we can naturally employ it to transfer across frames. Thus, we introduced Target Enhancement, including Temporal tokens for target feature aggregation, and Temporal Mamba for modeling these tokens. Specifically, after the first three stages, we provide Temporal Token $\boldsymbol{E}_{\boldsymbol{c}}^{\boldsymbol{0}}$ for each frame and insert them as $\boldsymbol{E}_{\boldsymbol{czx}}^{\boldsymbol{0}} \in \mathbb{R}^{[T \cdot (1+N_z) + (1+N_x)] \times D}$. The new sequence is fed into the last stage for additional target aggregation. After the final stage, we decompose the sequence $\boldsymbol{E}_{\boldsymbol{czx}}^{\boldsymbol{L}}$ into template $\boldsymbol{E}_{\boldsymbol{z}}^{\boldsymbol{L}}$, search $\boldsymbol{E}_{\boldsymbol{x}}^{\boldsymbol{L}}$ and their Temporal Tokens $\boldsymbol{E}_{\boldsymbol{c,z}}^{\boldsymbol{L}}$, $\boldsymbol{E}_{\boldsymbol{c,x}}^{\boldsymbol{L}}$ with highly aggregated target features. Next, these Temporal Tokens will be continued into Temporal Mamba, consisting of multiple Mamba Layers, to achieve temporal propagation across frames. Finally, the search features are refined by Temporal Token before passed into the box head. Follow-up experiments demonstrate the effectiveness of Target Enhancement and provide sufficient visualizations to illustrate its impact.

### 4.4 TRAINING AND INFERENCE

**Training.** The training processing of our TrackMamba generally follows current trackers (Yan et al., 2021; Cui et al., 2024) to train the whole tracking framework on the tracking datasets. We adopt the combination of $L_1$ loss and CIoU loss (Zheng et al., 2020) as follows:

$$L = \lambda_{L1} L_1(b_i, \hat{b}_i) + \lambda_{ciou} L_{ciou}(b_i, \hat{b}_i), \quad (6)$$

where $\lambda_{ciou} = 2$ and $\lambda_{L1} = 5$ are the trade-off weights of the combined loss, $b_i$ and $\hat{b}_i$ represent the ground-truth and the predicted box of the targets in search frames respectively.

Table 1: Comparsion on LaSOT (Fan et al., 2019), TrackingNet (Müller et al., 2018), and GOT-10k (Huang et al., 2021). The best two results are shown in red and blue fonts.

| Method | GOT-10k | | | TrackingNet | | | LaSOT | | |
|---|---|---|---|---|---|---|---|---|---|
| | AO(%) | $SR_{0.5}$(%) | $SR_{0.75}$(%) | AUC(%) | $P_{Norm}$(%) | P(%) | AUC(%) | $P_{Norm}$(%) | P(%) |
| SiamFC (Bertinetto et al., 2016) | 34.8 | 35.3 | 9.8 | 57.1 | 66.3 | 53.3 | 33.6 | 42.0 | 33.9 |
| DiMP (Danelljan et al., 2020) | 61.1 | 71.7 | 49.2 | 74.0 | 80.1 | 68.7 | 56.9 | 65.0 | 56.7 |
| SiamFC++ (Xu et al., 2020) | 59.5 | 69.5 | 47.9 | 75.4 | 80.0 | 70.5 | 54.4 | 62.3 | 54.7 |
| STMTracker (Fu et al., 2021) | 64.2 | 73.7 | 57.5 | 80.3 | 85.1 | 76.7 | 60.6 | 69.3 | 63.3 |
| TransT (Chen et al., 2021) | 67.1 | 76.8 | 60.9 | 81.4 | 86.7 | 80.3 | 64.9 | 73.8 | 69.0 |
| AutoMatch (Zhang et al., 2021) | 65.2 | 76.6 | 54.3 | 76.0 | - | 72.6 | 58.2 | - | 59.9 |
| KeepTrack (Mayer et al., 2021) | - | - | - | - | - | - | 67.1 | 77.2 | 70.2 |
| STARK (Yan et al., 2021) | 68.8 | 78.1 | 64.1 | 82.0 | 86.9 | - | 67.1 | 77.0 | - |
| MixCvT$_{256}$ (Cui et al., 2024) | 70.8 | 80.7 | 67.1 | 81.9 | 87.1 | 79.8 | 67.9 | 77.9 | 73.2 |
| MixViT$_{256}$ (Cui et al., 2024) | 69.7 | 78.9 | 66.4 | 82.3 | 87.7 | 80.6 | 68.0 | 78.0 | 73.7 |
| MixViT$_{384}$ (Cui et al., 2024) | 72.4 | 81.2 | 70.8 | 83.3 | 88.5 | 82.9 | 69.8 | **80.8** | 69.4 |
| **TrackMamba$_{256}$** | 70.9 | 80.8 | 67.5 | 82.9 | 87.6 | 81.2 | 69.7 | **79.7** | 74.6 |
| **TrackMamba$_{384}$** | **72.8** | **81.6** | **70.6** | **84.5** | **88.8** | **83.7** | **70.0** | 79.1 | **75.3** |
| **TrackMamba$_{512}$** | **74.0** | **82.7** | **71.0** | **84.7** | **88.5** | **84.0** | **70.1** | 78.8 | **75.1** |

**Inference.** During inference, we input $T$ templates, including static first frame and dynamic online templates, together with search region into TrackMamba to predict the target box. Since the target appearance varies in frames and it profoundly affects performance, we adopt the Score Prediction Module of MixFormer (Cui et al., 2024) to choose reliable online templates which produces the confidence score of prediction and selects the highest one when the update interval is reached. Note that we directly output the box prediction without any post-processing like the window penalty.

## 5 EXPERIMENTS

### 5.1 IMPLEMENTATION DETAILS

Our tracker is implemented in Python 3.10 using PyTorch 2.1.1. The models are trained on 8 NVIDIA A6000 GPUs and the inference speed is tested on a single NVIDIA A6000 GPU.

**Model.** MambaVision (Hatamizadeh & Kautz, 2024), a hybrid Mamba-Transformer model, adopted as the backbone with its ImageNet-1k (Deng et al., 2009) classification pretrain to initialize. The bounding box head is the corner-based head. Especially, since the features are various scales from the hierarchical backbone instead of the plain ViT, we modify the corner head as Fig. 3 for fusing the multi-scale output and refined features for more precise representation. Three variants with different input image pair resolutions of our TrackMamba are presented as follows:

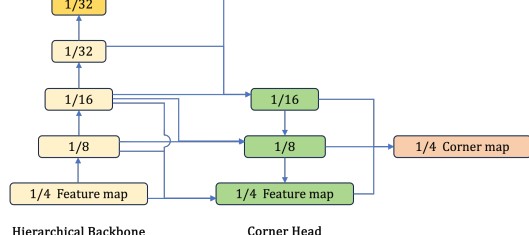

Figure 3: Modified Corner head to accept multiple scale feature from hierarchical backbone and the last feature refined by Temporal Token.

- TrackMamba-256. Template: 128×128 pixels; Search region: 256×256 pixels.
- TrackMamba-384. Template: 192×192 pixels; Search region: 384×384 pixels.
- TrackMamba-512. Template: 256×256 pixels; Search region: 512×512 pixels.

**Training.** In line with the traditional training datasets, our training data includes the training splits of LaSOT (Fan et al., 2019), GOT-10k (Huang et al., 2021), TrackingNet (Müller et al., 2018), and COCO (Lin et al., 2014) and the 1k forbidden sequences from GOT-10K training set are removed for fair comparison. As for GOT-10k test, we re-train our trackers with the GOT-10k train split following

Table 2: Ablation on different Scan Patterns, Direction and Interaction Modes. "Scan Pattern" means different rearrangement and flipping strategies mentioned in Section 4. "Interaction Mechanism" states whether to adopts the mechanism to implement the interaction or feature extraction only.

| # | Scan Pattern | bi-direction | Interaction Mechanism | | LaSOT | | | GOT-10k | | |
|---|---|---|---|---|---|---|---|---|---|---|
| | | | Mamba | Attention | AUC(%) | $P_{Norm}$(%) | P(%) | AO(%) | $SR_{0.5}$(%) | $SR_{0.75}$(%) |
| 1 | (a) | ✓ | ✓ | ✓ | 68.7 | 78.5 | 73.6 | 69.3 | 78.9 | 64.8 |
| 2 | (c) | ✓ | ✓ | ✓ | 53.1 | 56.4 | 51.4 | 59.5 | 67.2 | 50.1 |
| 3 | | | ✓ | ✓ | 67.9 | 77.3 | 72.7 | 68.1 | 77.2 | 63.6 |
| 4 | (b) | ✓ | ✓ | | 66.3 | 75.9 | 70.5 | 69.3 | 79.0 | 64.3 |
| 5 | | ✓ | | ✓ | 52.3 | 56.1 | 50.7 | 54.8 | 60.7 | 46.3 |
| 6 | (b) | ✓ | ✓ | ✓ | 69.7 | 79.7 | 74.6 | 70.9 | 80.8 | 67.5 |

Table 3: Ablation on Target Enhancement, including Scan Pattern (b) and (c) w/o Target Enhancement, insertion location of Temporal Token, and layer number of Temporal Mamba.

| # | Settings | | LaSOT | | | GOT-10k | | |
|---|---|---|---|---|---|---|---|---|
| | | | AUC(%) | $P_{Norm}$(%) | P(%) | AO(%) | $SR_{0.5}$(%) | $SR_{0.75}$(%) |
| 1 | w/o Temporal Token | Approach (c) | 54.1 | 58.4 | 52.5 | 59.6 | 66.8 | 50.9 |
| 2 | | Approach (b) | 66.9 | 75.7 | 70.8 | 66.2 | 74.5 | 61.7 |
| 3 | Temporal Token Location | Middle | 67.5 | 76.7 | 71.5 | 68.1 | 77.2 | 63.7 |
| 4 | | Tail | 68.9 | 78.5 | 73.4 | 67.6 | 76.8 | 63.4 |
| 5 | Temporal Mamba Layer | 1 Layer | 68.4 | 77.9 | 72.8 | 65.7 | 74.1 | 60.4 |
| 6 | | 2 Layer | 68.4 | 77.8 | 73.0 | 68.0 | 77.1 | 63.3 |
| 7 | Approach (b), Head, 3 Layer | | 69.7 | 79.7 | 74.6 | 70.9 | 80.8 | 67.5 |

its standard protocol. The training 500 epochs with 60k image pairs in each epoch, and each of 8 GPUs holds 32 image pairs. The network is optimized with the AdamW optimizer (Kingma & Ba, 2015) with weight decay of $1 \times 10^{-4}$. The initial learning rate of backbone is $4 \times 10^{-5}$ and $4 \times 10^{-4}$ of remaining modules, which dropped by a factor of 10 after 400 epochs. The data augmentations include the horizontal flip and brightness jittering.

**Inference.** The online template update interval and threshold are set to 200 and 0.5 by default, while selecting the template with the highest score from the Score Prediction Module. Following conventional process, the templates are target-center cropped and the search region is cropped from the current frame with the predicted target center position from the previous frame as the center.

### 5.2 COMPARISON WITH THE STATE-OF-THE-ART TRACKERS

To verify the performance of our proposed TrackMamba, we compare our evaluation results on several benchmarks, including LaSOT, TrackingNet, and GOT-10k. We focus our comparisons on representatives of the first generation of trackers, MixFormer (Cui et al., 2024), which adopt CvT (Wu et al., 2021) with ImageNet-22k classification pre-train or ViT (Dosovitskiy et al., 2021) with MAE (He et al., 2022) pre-train as its backbone, both of them are better pre-train than ours.

**GOT-10k.** GOT10k (Huang et al., 2021) is a large-resolution dataset with most 2K-resolution videos and its train and test splits are zero overlaps of object classes. Table 1 shows our tracker surpasses others at same resolution. Remarkably, at this large resolution benchmark, expanding the model input size resulted in a significant improvement, demonstrating the importance of input size.

**TrackingNet.** TrackingNet (Müller et al., 2018) contains 511 test sequences with diverse target classes. As shown in Table 1, our tracker benefits on diverse targets more than others.

**LaSOT.** LaSOT (Fan et al., 2019) is a long-term tracking benchmark containing 280 test videos. It shows our TrackMamba outperforms other trackers at same resolution while the poor performance gain from resolution increasing here is due to the low resolution of most of the videos in this dataset.

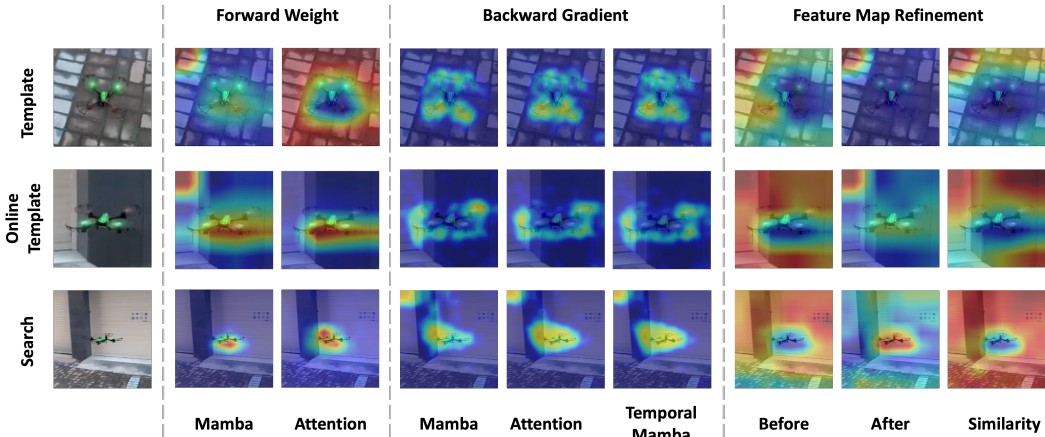

Figure 4: Visualization of the effect on Temporal Tokens. Each row indicates the impact of frame's Temporal Token on its, including: 1) the forward weight map of Mamba and Attention within backbone; 2) the backward gradient of Mamba and Attention within backbone and Temporal Mamba; 3) feature map before and after refinement with Temporal Token and similarity of refinement.

Table 4: Ablation on the various number of template frames (including the first frame).

| Template | LaSOT | | | GOT-10k | | | FLOPs(G) |
|---|---|---|---|---|---|---|---|
| Number | AUC(%) | $P_{Norm}$(%) | P(%) | AO(%) | $SR_{0.5}$(%) | $SR_{0.75}$(%) | |
| 1 | 68.0 | 77.6 | 72.9 | 68.1 | 77.0 | 63.7 | 73.28 |
| 2 | 69.7 | 79.7 | 74.6 | 71.0 | 81.0 | 68.0 | 83.29 |
| 3 | 66.6 | 75.3 | 70.4 | 66.5 | 75.2 | 61.4 | 93.25 |

## 5.3 ABLATION AND ANALYSIS

In this section, we give a thorough analysis on the TrackMamba-256 model trained on four tracking datasets, and perform detailed ablation studies on LaSOT and GOT-10k benchmarks.

**Study on Scan Patterns.** The arrangement order and scanning manner of the frame tokens play a key role in intra- and inter-frame modeling. To verify the validity, as shown in Table 2, we experimented with different scan patterns. First, bi-directional scanning (#3 v.s. #6) expands receptive field, which is more convenient for extraction, interaction, and aggregation. Compared to the whole flipping in sequential-rearrange-whole-flip (#1), the separate inversion in sequential-rearrange-separate-flip (#6) ensures the target information flow and avoids the distraction from search region. Unfortunately, even if the cropping strategy ensures the center location, interleaved-rearrange-whole-flip (#2) still hardly guarantees that target exists in the same position across frames, and the repeated cross-frame scanning may break the intra-frame continuity for feature extraction.

**Study on Interaction Modes.** Since the hybrid model adopt both Mamba and Attention mechanisms, for more obvious comparison of interaction ability, we switch one of them to feature extraction only respectively. As shown in Table 2, in addition to the optimal performance achieved by employing both, Mamba-only achieves better performance than Attention-only (#4 v.s. #5), demonstrated that the quality long sequence modeling capability of Mamba transfer target information with hidden state well enough. It suggests Mamba is a stronger alternative to Attention while Attention could assist back during discontinuous tokens interaction, which is a good use of the hybrid model.

**Study on Target Enhancement.** As shown in Table 3, we ablate the impact of the Target Enhancement. First, we remove the it (#2 v.s. #7), the performance degradation demonstrates it achieving target feature aggregation and propagation simply but efficiently. Next, we explored different insertion locations for the token, including the head(#7), middle(#3), and tail(#4) of each frame, showing the head location could aggregate representation better than the others. Furthermore, we attempted different layer numbers of Temporal Mamba, experiments #5-7 show that more layers achieve better performance, indicating the effectiveness of transferring aggregated features.

**Study on the various number of template frames.**    With more available templates, search region receives target representation at various moments for robust tracking. As shown in Table 4, with one more template, the performance growth shows the ability to model more than one representation. Not as expected, more templates lead to worse performance. We analyze that Mamba does not directly interact with discontinuous tokens so that the best feature in first template will be interfered while the hidden state is difficult to carry more information well with its limited dimension.

## 5.4 Visualization

**Visualizations on the effect of Temporal Token.**    To illustrate the role of the token, we visualize its impact in Fig. 4. The first two columns represent the effect of each token on its frame in Mamba and Attention in the forward, while the next two columns represent the gradient collected in the backward. It can be noticed that the impact region focuses on the target center during forward to collection, while on the edges during backward to location. The last three columns show the search refinement with Temporal Token, showing that the feature map focuses on the target after refinement with tokens, indicating that one simple but effective token collects enough features of the target.

**Visualizations on effective receptive field.**    To explore the effective receptive fields (Luo et al., 2016) across various frames, which measures the relevance of input to output within the model, we present a comparative analysis for intra- and inter-frame modeling. Specifically, given a central area in each frame, we visualized the corresponding receptive. As shown in Fig. 5, the first and second columns represent the ERF of center area in two templates from themselves, and the remaining columns represent them in search frames from all frames. It is significant to see those areas exhibit fixed local ERF before training, and

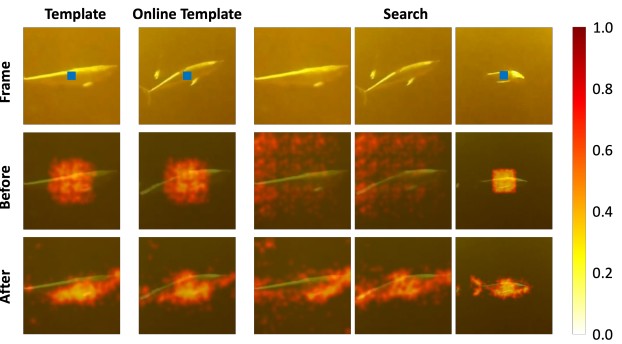

Figure 5: Visualizations of Effective Receptive Field (ERF) of blue area before and after training. The first and second columns represent the areas of templates affected by themselves, while the subsequent columns represent the areas of search frames that are affected from all frames.

more importantly, template frames have a pretty sparse ERF for the search, indicating the modeling ability is inadequate at this point, failing to transfer information. After training, the ERF of the intra-frame becomes more fit the shape and search region dynamically locating the corresponding field from templates, showing excellent modeling ability, which can be naturally used to transfer target information in tracking tasks.

## 5.5 Limintions and Future

The most serious problem with current framework is no performance improvement with more templates. On the one hand, we consider applying the new proposed Mamba2 (Dao & Gu, 2024) to our framework for better carrying. On the other hand, we could modify the transfer path, such as adding scanning branches from the first frame to implement a longer temporal tracker in the future.

## 6 Conclusion

This work proposes TrackMamba, a Mamba-Transformer tracking framework based on Track-Mamba Blocks with various scan patterns and Attention Blocks, aiming to transfer target feature with the scanning mechanism of Mamba. By leveraging additional Target Enhancement with Temporal token and Temporal Mamba, it obtains target aggregation and high representation transferring across frames directly. Extensive experiments the proposed tracker beats the first-generation one-stream Transformer-based tracker at same resolution on performance and memory consumption, especially in terms of scalability at large resolutions. We expect this work can catalyze more compelling research to Mamba-based tracker on large resolution and long sequence tracking.

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

## A APPENDIX

### A.1 ANALYSIS ON EFFICIENCY AND PERFORMANCE

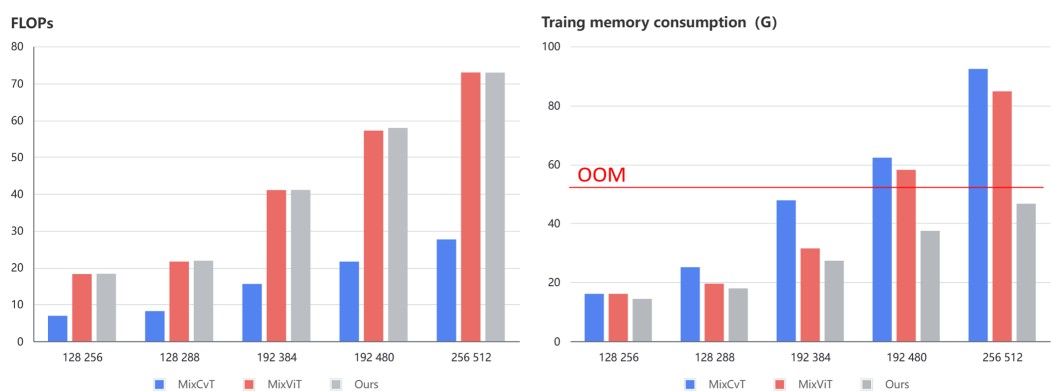

Figure 6: Comparison FLOPs and training Memory

Fig. 6 shows the FLOPs and training memory consumption of MixCvT (Cui et al., 2024), MixViT (Cui et al., 2024), and our method with different input resolutions. In the FLOPs chart, we observe that our computational cost is close to that of MixViT, indicating that our model does not suffer from significant inefficiency due to the lack of optimization of the new architecture. Additionally, we tested the memory consumption during training, and the results show that when the input image resolution comes to 256 and 512, it brings significant computation burden to both MixCvT and MixViT, while our structure significantly reduces memory consumption, alleviating the memory pressure at high resolutions. This demonstrates that our model consistently achieves a balance between accuracy and efficiency.

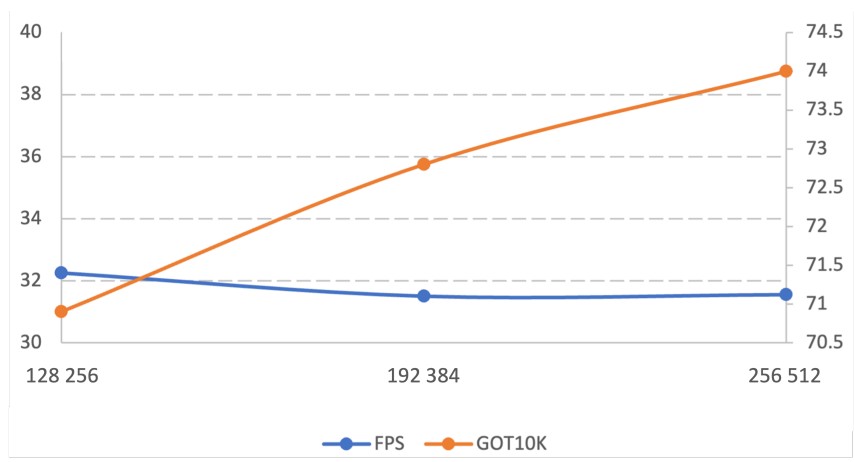

Figure 7: Comparison FLOPs and training Memory

The GOT-10k serves as a large-scale benchmark, providing a large number of high-resolution videos. However, A can only work at low resolutions due to quadratic computation consumption limitations, resulting in inadequate performance. We measured speed and GOT-10k performance of Track-Mamba at different resolutions. As shown in Fig 7, with increasing input resolution, the tracker's performance on GOT-10k (Huang et al., 2021) improves dramatically with only a small decrease in efficiency, demonstrating its excellent scalability on resolution.

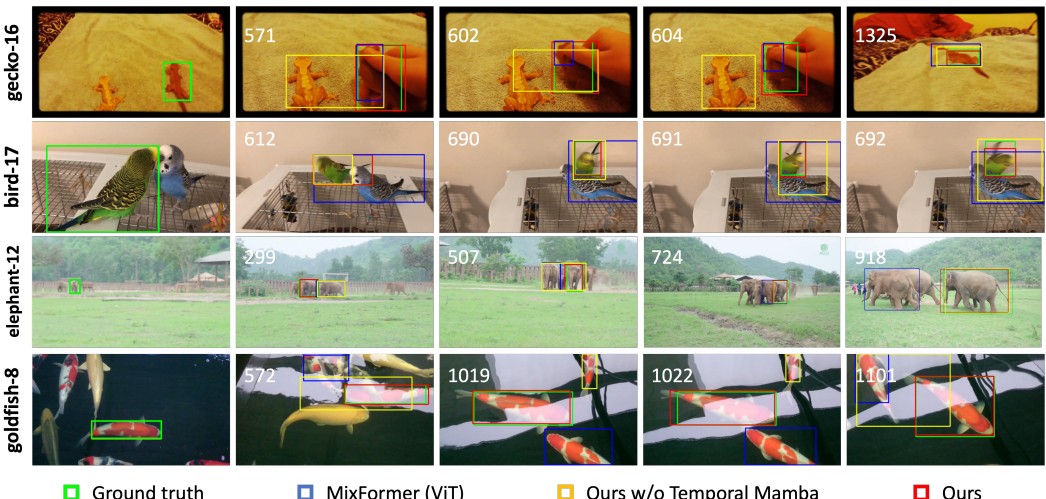

Figure 8: Comparison tracking results on LaSOT benchmark.

## A.2 QUALITATIVE COMPARISON

Fig. 8 displays the qualitative comparision of our method with MixFormer (Cui et al., 2024). As shown by gecko-16 sequence, our method demonstrates superior performance with similar target and background. In bird-17 and elephant-12 sequence, our design results in better performance under multiple objects and serious occlusion while MixFormer and our model without Temporal Mamba tend to drift to other objects. Additionally, goldfish-8 sequence, there remains the problem of changing appearance due to reflections caused by the target being underwater, the others struggle to locate the target whereas ours does. These findings demonstrate the effectiveness of the proposed method in dealing with various challenges of tracking.

