# OpenReview forum: "TrackMamba: Mamba-Transformer Tracking"
_ICLR.cc/2025/Conference — ICLR 2025 Conference Withdrawn Submission_

### Official Review · Reviewer_PjTd · 2024-10-18

**Soundness:** 3
**Presentation:** 4
**Contribution:** 3
**Rating:** 5
**Confidence:** 4

**Summary:**

The author applies a Mamba-based backbone to the one-stream visual tracking framework. The author explores various scan patterns among the template and search frames for this Mamba based tracker. Other widely-applied tricks, such as dynamic templates and motion tokens, are also applied to the tracker. The experiments show that TrackMamba outperforms Transformers at the same resolution while reducing memory usage with larger resolutions.

**Strengths:**

1. This paper has an interesting topic, which applies the Mamba-based structure to the visual tracking framework. The motivation also makes sense as the Mamba can have linear computational complexity to the transformer-based architecture.

2. The overall framework is neat, though some widely applied tricks, such as motion tokens to strengthen the temporal modeling, are applied. The Mamba-based tracking framework has the potential to contribute to developing next-generation trackers.

3. The experiments are also extensive. The benchmark results show some SOTA performance under specific settings. The author also strengthens the computation flops when the input resolution is larger, which validates the strength of Mamba-based models.

**Weaknesses:**

1. I am most concerned that the application of the Mamba-based backbone does not show enough strengths. Firstly, many widely adopted tricks from transformer-based one-stream trackers, such as dynamic templates and motion tokens, are applied. The author should develop the specific modules or functions under the long-sequence modeling framework (mamba)， such as more templates. The larger input resolution setting is good, but the speed of the method is around 30FPS, which is far lag behind the transformer trackers (typically larger than 80FPS). Thus, though the computation flops are reduced, the inference speed is not competitive. Thus, it makes the proposed tracker not as competitive as the 2024 year sota trackers in various benchmarks.

2. The second point is that the overall backbone network is borrowed from the MambaVision paper, which was developed for general vision tasks as a backbone network. The difference only lies in the scanning pattern of the template and search images.  The author's exploration of the scanning pattern is straightforward and only involves the order of the template and search frames. It lacks enough insights, and I think the casual modeling of the Mamba should be explored for this video task.

3. As addressed by point 1, the performance of the method is not competitive, removing the additional tricks. Also the speed
(30fps) which is important for tracking, is really not satisfying as the pipeline is generally neat and simple.

**Questions:**

Please see the weakness. I am glad to see the new Mamba model applied in tracking, and I do not emphasize SOTA performance too much. The reason I did not give a positive review at the beginning is that the explorations on Mamba were not adequate. However, I would raise the rating if my concerns are addressed.

---

### Official Review · Reviewer_Ymzw · 2024-10-23

**Soundness:** 2
**Presentation:** 2
**Contribution:** 2
**Rating:** 3
**Confidence:** 4

**Summary:**

The manuscript presents a novel target tracking method called TrackMamba, which employs a hybrid Mamba-Transformer model as its backbone. TrackMamba aims to address the issue of high memory consumption in existing single-stream Transformer-based trackers when dealing with high-resolution and long sequences. The main contributions are as follows:

●	This manuscript introduces the TrackMamba tracking framework, alleviating computational burdens while ensuring high accuracy.
●	This manuscript designs various scan patterns to better arrange and flip the input sequences within the TrackingMamba Block.
●	This manuscript proposes a Target Enhancement module that utilizes Temporal Tokens and Temporal Mamba to enable effective information propagation across frames, enhancing the aggregation and transmission of target information.

**Strengths:**

1.	TrackMamba introduces several novel Mamba Scan Patterns for better scanning of input sequences. Through the scanning mechanism of Mamba, it successfully achieves the transfer of target features and employs target enhancement techniques to enable high representation transfer across frames.
2.	TrackMamba combines tracking efficiency with performance. In comparisons with existing state-of-the-art trackers, including FLOPs and trainingmemory consumption, TrackMamba significantly reduces training memory consumption without decreasing FLOPs, while maintaining high tracking accuracy.
3.	For the first time, TrackMamba effectively integrates Temporal Tokens and Temporal Mamba alongside Template Tokens and Search Tokens, enabling efficient aggregation of target features and propagation of information across frames.
4.	The illustrations in the manuscript are concise and clear, effectively conveying the construction of TrackMamba and the design concepts behind the three Scan Patterns.

**Weaknesses:**

1.	1.	Although this manuscript proposes a novel Mamba-based tracker with great performance both on tracking accuracy and memory consumption, it lacks the comparisons with the existing representative Mamba-based trackers, such as Mamba-FETrack[1] and so on.
[1] Huang J, Wang S, Wang S, et al. Mamba-FETrack: Frame-Event Tracking via State Space Model[J]. arXiv preprint arXiv:2404.18174, 2024.
2.	The paper does not provide a detailed explanation of the Enhancement design within the Target Enhancement module(in Figure 1). Specifically, how is the Search Feature refined by the Temporal Token through the Enhancement? I advise providing an extra illustration or pseudocode for how the Temporal Token refines the Search Feature.
3.	In the comparison experiments of Scan Patterns (in Table 2), the results for sequential-rearrange-whole-flip (#1 in Table 2) far exceed those for interleaved rearrange-whole-flip (#2 in Table 2). However, based on the narrative logic of the manuscript, the interleaved rearrange-whole-flip method should be superior to sequential-rearrange-whole-flip (as discussed in lines 281-289 of the manuscript and section 5.3 as well as in Table 2). I suggest you provide a more in-depth analysis of why this phenomenon exists and thus uncover its potential implications for the proposed method.

**Questions:**

1.	Although TrackMamba performs well at high resolutions, the performance improvement in low-resolution sequences (LaSOT, in Table 1) is not significant. This may limit its generalizability across videos of different resolutions.
2.	With more available templates, the search region is expected to receive target representation at various moments for robust tracking. However, the current framework shows no performance improvement with the addition of more templates (Table 4).
3.	Although TrackMamba aims to reduce memory consumption, computational complexity may still become a problem as input resolution increases. For resource constrained environments, this can be a challenge.

---

### Official Review · Reviewer_FQeG · 2024-11-01

**Soundness:** 2
**Presentation:** 3
**Contribution:** 2
**Rating:** 5
**Confidence:** 3

**Summary:**

This paper introduces a MambaTransformer-based tracker tailored for robust tracking performance. To enhance the Mamba-based Transformer for tracking, the authors devise TrackMamba Blocks and Attention Blocks to address the limited hidden state problem. The method is formulated well. Promising results are presented.

**Strengths:**

This paper justifies the feasibility of applying Mamba to tracking. Several useful formulations and components are designed to achieve effective tracking result.

**Weaknesses:**

In general, I think this paper needs to provide more solid justifications for their overall contributions. Please see my questions below.

**Questions:**

(1) The paper's motivation needs more explanations. There are many aspects that confuse me. For example, in line 067, "rearrangement and flipping of input sequence", it is not clear to me what is this problem. This is the first time introducing this statement but without comprehensive explanations. From the rest of the paper, I can guess that this statement is related to the formulation when applying Mamba to the tracking problem. If this is true, please add more discussions on this.

(2) Mamba is increasingly popular these days, but it is still necessary to demonstrate the necessity of applying Mamba to the tracking problem. Is the Mamba better at visual modelling? Or is it better at memorizing target historical appearances? Throughout the paper, it remains unclear to me why Mamba is an important future direction in tracking. Maybe, Mamba is good at balancing between short-term appearance updates and long-term appearance updates like in [1]? I think the authors need to discuss this in their paper.

(3) I have doubts about the experimental results. I do not think the compared methods represent state-of-the-art performance, e.g. the performance of [2][3]. Although it may not be necessary to outperform these SOTA methods, at least the authors should discuss them and explain clearly in what aspects their method has advantages over the existing method.

I am open to discuss with the authors before making my final decision.

[1] Hong, Zhibin, et al. "Multi-store tracker (muster): A cognitive psychology inspired approach to object tracking." Proceedings of the IEEE conference on computer vision and pattern recognition. 2015.
[2] Xu, Yuanyou, Zongxin Yang, and Yi Yang. "Integrating boxes and masks: A multi-object framework for unified visual tracking and segmentation." Proceedings of the IEEE/CVF International Conference on Computer Vision. 2023.
[3] Bai, Yifan, et al. "Artrackv2: Prompting autoregressive tracker where to look and how to describe." Proceedings of the IEEE/CVF Conference on Computer Vision and Pattern Recognition. 2024.

---

### Note · Authors · 2024-11-14

I have read and agree with the venue's withdrawal policy on behalf of myself and my co-authors.